# Coupling Semi-supervised Learning with Reinforcement Learning for Better Decision Making — An application to Cryo-EM Data Collection

**Ziping Xu**[*]
Department of Statistics
Harvard University

**Quanfu Fan**
Amazon

**Yilai Li**
Life Sciences Institute
University of Michigan

**Emma Rose Lee**
Department of Biology
Massachusetts Institute of Technology

**John Cohn**
MIT-IBM Watson AI Lab

**Ambuj Tewari**
Department of Statistics
University of Michigan

**Seychelle Vos**
Department of Biology
Massachusetts Institute of Technology

**Michael Cianfrocco**
Life Sciences Institute and
Department of Biological Chemistry
University of Michigan

## Abstract

We consider a semi-supervised Reinforcement Learning (RL) approach that takes inputs from a perception model. Performance of such an approach can be significantly limited by the quality of the perception model in the low labeled data regime. We propose a novel iterative framework that simultaneously couples and improves the training of both RL and the perception model. The perception model takes pseudo labels generated from the trajectories of a trained RL agent believing that the decision-model can correct errors made by the perception model. We apply the framework to cryo-electron microscopy (cryo-EM) data collection, whose goal is to find as many high-quality micrographs taken by cryo-electron microscopy as possible by navigating at different magnification levels. Our proposed method significantly outperforms various baseline methods in terms of both RL rewards and the accuracy of the perception model. We further provide some theoretical insights into the benefits of coupling the decision model and the perception model by showing that RL-generated pseudo labels are biased towards localization which aligns with the underlying data generating mechanism. Our iterative framework that couples both sides of the semi-supervised RL can be applied to a wide range of sequential decision-making tasks when the labeled data is limited.

## 1 Introduction

Decoupling representation learning or perception learning from Reinforcement Learning (RL) is commonly used to improve performance in RL applications (Stooke et al., 2021). For example, the idea of state abstraction for RL concerns learning a low dimensional state representation to deal with a large state space (Jong and Stone, 2005; Abel et al., 2016; Raffin et al., 2018; Ho, 2019). The success of decoupling perception learning from RL depends on the quality of the perception model, which often requires a large amount of labeled data for training. In many realistic scenarios,

---

[*]This work was doen when Ziping was an intern at IBM.

acquiring fully labeled datasets is nevertheless costly and sometimes infeasible, while acquisition of unlabeled data is relatively inexpensive. Such situations render semi-supervised learning (SSL) (Zhu, 2005) a natural choice for obtaining good perception representations with limited annotations for RL. However, a naive application of SSL to perception models may not necessarily lead to promising results for RL because a) the improvement of SSL in the case of a small number of labeled data can be too subtle to facilitate RL; and b) the improved overall accuracy of the perception model may not be directly relevant to better RL policies.

Interestingly, in many cases, an RL agent can provide useful feedback to the perception model through the quality of sampled trajectories during learning. We investigate the idea of improving perception modeling by RL under an SSL setting with limited labeled data and vice versa. We specifically consider a family of navigation problems with the goal of discovering as many targets of interest as possible. For example, Scavenger hunt (Yedidsion et al., 2021) trains a robot to search places with high probability of finding the targets, and Fan et al. (2022) applies RL to optimize microscope movement for efficient CryoEM data collection. The structure of such navigation problem permits a straightforward approach to generates pseudo labels directly from current policies' rollouts and correct mistakes made by the perception model, as illustrated by the example in Figure 1.

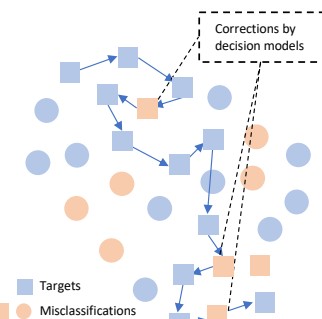

Figure 1: A toy example of path planning, where the goal of the agent is to find as many targets (squares) of interest as possible for a given moving distance. A pretrained model is used to separate targets from others (circle). Illustrated is a trajectory planned by the agent, along which some misclassified squares (orange) by the model are corrected.

In light of the intuition above, we propose to couple perception modeling and RL in an iterative framework to mutually enhance each other in scenarios with label shortage issues. Specifically, the RL agent gives helpful feedback to the perception model by generating pseudo labels through a learned RL policy. The resulting improved perception representations, which, in turn, provide better input to RL, lead to more effective RL policies. We alternate perception modeling and policy learning iteratively to refine them for multiple rounds. Since both perception modeling and RL use labeled and unlabeled data for training, we dub our approach *SSL²-RL* (SSL for both RL and perception learning).

SSL has been applied to improve RL (SSL-RL) where the reward function can only be evaluated in some settings but not all. For instance, Finn et al. (2016) uses unlabeled trajectories for a better importance sampling estimator of a particular parameter in the entropy objective function. Konyushkova et al. (2020) learns a reward function to annotate the trajectories without generating new unlabeled trajectories to improve the reward function. Fu et al. (2017) learns a discrimination model to discriminate the RL trajectories from the positive examples in a binary reward setup. The major difference between our approach and SSL-RL is that SSL-RL studies how to enable better reward modeling with unlabeled data while our approach focuses on exploring mutual benefits of combining perception learning and RL for better decision making. To the best of our knowledge, the latter has not been studied previously.

## 1.1 Related literature

**Label Propagation.** Label propagation propagates labels through a dataset along high density areas defined by unlabeled data. It follows the intuition that close points should have similar labels. Zhu and Ghahramani (2002) iteratively propagates labels using a linear combination of adjacent nodes defined on a graph. Su et al. (2015); Vernaza and Chandraker (2017); Jabri et al. (2020) perform label propagation through random walk. Our method can be seen as a special way of propagating labels through a decision-making models, which incorporates both context information and the geometric information. Cai et al. (2021) proposed to optimize the loss with a regularization on the inconsistency over samples within the same neighborhood.

**Semi-supervised RL.** Semi-supervised RL concerns the problem where the agent must perform RL when the reward function is known in some settings, but cannot be evaluated in others. For semi-supervised RL, a wide range of pseudo reward is generated. For example, Finn et al. (2016); Fu et al. (2018); Singh et al. (2019); Konyushkova et al. (2020) learns a classifier for reward labeling using a labeled dataset, which is applied to optimize a entropy-regularized objective for an unlabeled dataset. Yu et al. (2022) states that a zero pseudo reward is sufficient for tasks using sparse reward functions. Some other pseudo rewards are proposed using task-specific prior knowledge such as distance to goals in goal-conditioned settings (Andrychowicz et al., 2017). In contrast to others, some approaches directly imitate expert trajectories to achieve high-levels of performance without requiring reward labels (Ross and Bagnell, 2012; Ho and Ermon, 2016).

**Cryo-EM Data Collection.** We have focused this work on addressing the issue of cryo-EM data collection. Cryo-EM serves as a critical tool for determining the three-dimensional structures of biological macromolecules. As such, cryo-EM is a powerful tool in the development of vaccines and therapeutics to combat diseases such as COVID-19. Within weeks of the release of the genomic sequence of SARS-CoV-2, cryo-EM determined the first SARS-CoV-2 spike protein structure (Wrapp et al., 2020). Since this original publication, cryo-EM was used to determine additional SARS-CoV-2 structures such as spike protein bound to antibody fragments (Lempp et al., 2021; Scheid et al., 2021), remdesivir bound to SARS-CoV-2 RNA-dependent RNA polymerase (Bravo et al., 2021; Yin et al., 2020; Kokic et al., 2021), and reconstructions of intact SARS-CoV-2 virions (Yao et al., 2020; Ke et al., 2020).

## 2 Problem formulation

We study the problem of training RL policies for navigation, where there are a large amount of unlabeled data whereas very few labels are available for learning perception representations for RL. We start by defining the RL environment, which consists of four major elements, i.e the state space, action space, transition function and reward function. The state space is a set of tuples, where each state is denoted by $(s, x, y)$, where $s \in \mathcal{S}$ encodes the context information that uniquely identifies each state, $y \in \{0, 1\}$ is a binary label and, $x \in \mathcal{X}$ is the input feature that can be used to predict $y$. Depending on the actual application, the context can be interpreted as the geometric information that encodes the location of the state on a map. Whenever is clear from the context, we let $y(s)$ be the true label corresponding to the state $s$. At the step $t$ the agent is provided with an action set $\mathcal{A}_t$ and the next reward and state are sampled from the $R : \mathcal{S} \times \mathcal{A} \mapsto [0, 1]$ and $T : \mathcal{S} \times \mathcal{A} \mapsto \mathcal{S}$. We consider a deterministic transition function. Throughout the paper, we consider a reward function that is directly relevant to the labels, i.e., $R(s, a) = \mathbb{1}(y(s) = 1) + c(s, T(s, a))$, where $c : \mathcal{S} \times \mathcal{S} \mapsto \mathbb{R}$ is a cost function. In the context of navigation problem, $c$ prevents the agent from conducting large movements.

We consider a semi-supervised learning scenario for both the perception model and Reinforcement Learning. We are given both labeled and unlabeled dataset denoted by $\mathcal{L} = \{s_i, x_i, y_i\}_{i=1}^{N_L}$ and $\mathcal{U} = \{s_i, x_i\}_{i=1}^{N_U}$, respectively, where $N_L$ and $N_U$ are their sizes. The perception model is a mapping $f : \mathcal{X} \mapsto [0, 1]$ that predicts the positive label probability with input feature $x$. Note that we consider a binary label for easier presentation, but our framework can be extended to the multi-class case.

### 2.1 Cryo-EM Data Collection

Cryo-EM is a key technique for structural biology that enables 3D structure determination of important macromolecular complexes and membrane proteins Wrapp et al. (2020). Cryo-EM data collection involves steering transmission electron microscopes hierarchically at different magnification levels (as shown in Figure 2) to explore a grid with the goal of identifying and collecting high-quality micrographs at high magnification. This sequential process includes several mechanical operations to allow microscope navigation to different regions of a grid, namely grid switching, square switching, and patch switching. An effective data collection session aims at finding a sequence of holes where there is a considerable portion of high-quality micrographs. However, it is a highly involved and time-consuming process that requires expertise and skills to make decisions at different levels of microscope operations.

To mitigate the inefficiency in cryoEM data collection, Fan et al. (2022) proposed to train an RL agent to automate this process. Their framework is called cryoRL, which first trains an image classifier.

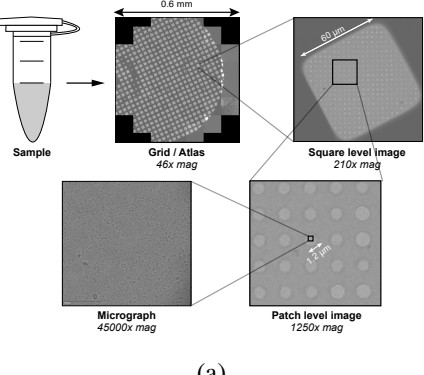

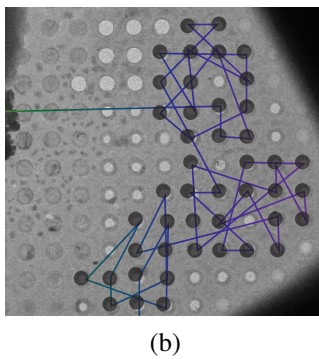

(a)

(b)

Figure 2: (a) (Figure 2 in Fan et al. (2022)) Overview of cryo-EM data collection. A purified sample is prepared and vitrified on the support grid. The atlas image provides a low magnification overview by stitching multiple "grid-level" images into a single montage. Next, users will select specific squares to image at medium magnification. After inspection, the user selects "patch" areas on the square to inspect holes with higher magnification, using the patch image to decide on holes to collect for micrographs. The micrographs contain high-resolution images for downstream data processing. (b) (Adapted from Li et al. (2022)) A trajectory collected by a trained RL policy (only for illustrative purpose).

The predictions of the image classifier as well as the distributions of labels within each grid, square, patch are used as features to train a DQN agent. They maximize the total number of high-quality holes within a fixed budget of time, which cast a requirement on an efficient path that does not need to steer the microscopes too frequently. A similar practice can also be found in Li et al. (2022).

CTF (contrast transfer function) is used to measure the quality of a hole. As a variable evaluated on multiple micrographs for each hole, it is infeasible to obtain CTFs for all the holes in a dataset before navigation. In practice, while a numerous number of samples are generated in daily data collection, only a small proportion of them can be actually evaluated and labeled, which makes cryo-EM data collection a perfect example to apply SSL approaches.

To fit cryo-EM data collection into the proposed problem formulation, we let each hole in the cryo-EM data collection be a state in the problem state. A hole can be represented as $\{s_i, x_i, y_i\}$, where $s_i = (\text{grid}_i, \text{square}_i, \text{patch}_i)$ represents grid, square and patch indices of the $i$-th hole. $x_i$ is the hole-level image of the $i$-th hole and $y_i$ represents the true quality of the hole.

## 3 Proposed method

We first discuss the training of the perception model and the RL model separately, before the iterative algorithm that integrates the two models are presented.

**Perception Model Training.** The quality of the perception model determines the overall quality of the RL agent. We propose to use the semi-supervised learning method, i.e. *FixMatch* (Sohn et al., 2020). *FixMatch* adds an unsupervised loss that regularizes the inconsistency between the strongly augmented and weakly augmented inputs from the unlabeled dataset. Recall that $f : \mathcal{X} \mapsto [0, 1]$ is the perception model. We let $P_f : \mathcal{X} \mapsto [0, 1]^2$ be the predicted label distribution over $\{0, 1\}$. The unsupervised loss is given by

$$l_U(f) = \sum_{i=1}^{N_U} \mathbb{1}(\max\{P_f(x_i^w)\} \geq \tau) H\left(P_f(x_i^w), P_f(x_i^s)\right), \tag{1}$$

where $x_i^w, x_i^s$ are the weakly and strongly augmented inputs of the $i$-th input in unlabeled data, and $H$ is the entropy function between two distributions.

In the cryo-EM task, we solve a binary image-classification problem, using a CTF threshold 6. As seen in Table 1b), the performance of a supervised model trained from the fully labeled dataset is $\sim 65\%$ only, indicating the classification task is nontrivial. As shown in Fig. 6 of the Appendix C,

with a cutoff threshold 6.0, many samples in the hole data lie around the threshold, suggesting that the training data is quite ambiguous.

**RL Policy Training.** Since the labeled data can be highly limited and the training of RL is unstable with a small number of observations, we train RL on both labeled and unlabeled data to utilize the information from the unlabeled data. As the pretrained classifier can be seen as a prediction on the reward function (without movement cost), it is natural to follow the commonly used approach that generates pseudo rewards through predicted reward labels (Finn et al., 2016). Let the reward at step $t$ be $\tilde{r}_t = y_{t+1}\mathbb{1}(s_{t+1} \in \mathcal{L}) + f(x_{t+1})\mathbf{1}(s_{t+1} \in \mathcal{U}) - c(s_t, s_{t+1})$. For instance, in cryo-EM task, whenever a hole in $\mathcal{L}$ is visited, a reward is generated from the true labels. If the hole is unlabeled, a pseudo reward is given by the predicted probability of being low CTF. Following Fan et al. (2022), we add to the final rewards an extra cost function that penalizes large movements (See Appendix B).

For cryo-EM task, we define a duration function $\tau : \mathcal{S} \times \mathcal{S} \mapsto \mathbb{R}$, which measures the amount of time it takes to move from one state to the other. The duration is directly associated with the movement cost of the microscopy. We consider a constrained RL, which terminates an episode whenever the cumulative duration $\tau(s_t, s_{t+1})$ reaches a threshold $\tau_{max}$. With the pseudo rewards, we train a regular offline DQN on the whole dataset (Van Hasselt et al., 2016) to optimize the following objective function:

$$\max \sum_{l=0}^{n_l} \tilde{r}_l \quad \text{s.t.} \sum_{l=0}^{n_l} \tau(s_t, s_{t+1}) \leq \tau_{max}, \text{ where } n_l \text{ is the index when terminated.}$$

**Pseudo Labels for Perception Models.** Since we consider a navigation problem, where a visit on a trajectory directly indicates the chance of finding a target of interest, we generate pseudo labels straightly from the visiting orders of trajectories as opposed to other SSL-RL methods (Finn et al., 2016; Fu et al., 2017). This approach allows us to back-propagate the geometric structural bias learned by the RL agent to the perception model. Let $(S_1, \ldots, S_{N_U+N_L})$ be the sequence of states the policy iterates until all the states are visited. For a given cutoff $N_C > 0$, we label the first $N_C$ states as positive while the rest of states negative. Note although we evaluate on the whole dataset, we will only use the pseudo labels for unlabeled data. A visualization of the process can be found in the right panel of Figure 3. As the starting point is chosen in a stochastic way, we evaluate the policy for $M$ independent times, which gives $M$ pseudo labels for each state for a more robust labeling process. Let the pseudo labels for the $i$-th state in the $m$-th run be $\bar{Y}_{im}$. Let the pseudo label $\bar{Y}_i$ be the majority of $\{\bar{Y}_{i1}, \ldots, \bar{Y}_{im}\}$. We let the confidence of each pseudo label be $p_i = \sum_{m=1}^{M} \bar{Y}_{im}/M$ if $\bar{Y}_i = 1$ and $1 - \sum_{m=1}^{M} \bar{Y}_{im}/M$ if $\bar{Y}_i = 0$.

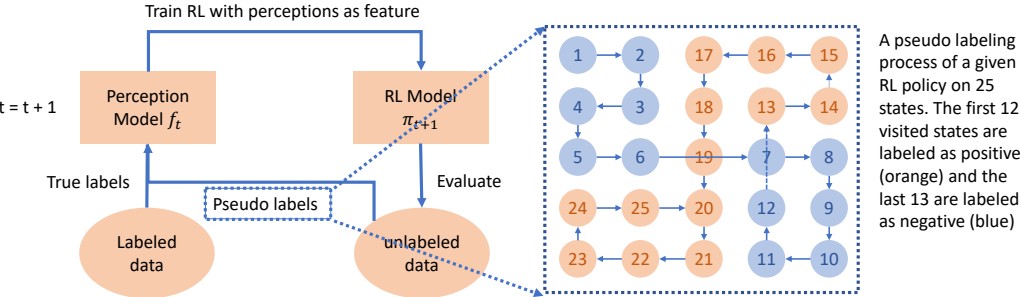

Figure 3: An iterative semi-supervised framework for perception and RL models. On the round $t$, the framework trains a RL agent $\pi_{t+1}$ that takes the perception model $f_t$ as input. By evaluating the agent $t + 1$ on the unlabeled data, it generates a pseudo label for each visited state. The perception model at the next step is trained on both labeled data and unlabeled data with pseudo labels.

**Iterative Framework.** Our main idea is to integrate RL and perception learning. We propose to feed the pseudo labels back to the perception model. We fine-tune the pretrained classifier on the whole dataset using both pseudo labels and true labels available. Each input in the unlabeled data is sampled with a probability proportional to its confidence. Let CE : $[0, 1]^2 \times \{0, 1\} \mapsto \mathbb{R}$ be the cross entropy loss. The loss function of fine-tuning the perception model with soft pseudo labels from RL

trajectories is then given by $l(f) = l_{\text{sup}}(f) + \lambda l_{\mathcal{U}}(f)$, where

$$l_{\text{sup}} = \sum_{i=1}^{N_U + N_L} p_i \, \text{CE}(f(x_i), \bar{Y}_i) \mathbb{1}(s_i \in \mathcal{U}) + \text{CE}(f(x_i), y_i) \mathbb{1}(s_i \in \mathcal{L}).$$

An overview of our propose method is given in Algorithm 1. The approach may not reach the best performance in a single round. Thus, we repeat the process multiple times, and select the model with the best validation performance as the final output.

---

**Algorithm 1** Iterative framework for the joint training of the perception and RL models

---

**Input:** Labeled and unlabeled dataset $\mathcal{L}, \mathcal{U}$ and the number of iterations $\mathcal{K}$
Pretrain teacher classifier $f_0$ on $\mathcal{L}$ and $\mathcal{U}$ using *FixMatch* .
**for** $t = 1, \ldots \mathcal{K}$ **do**
    Train RL on both $\mathcal{L}$ and $\mathcal{U}$ with pseudo rewards predicted by classifier $f_{t-1}$.
    Generate pseudo labels $\bar{Y}_1, \ldots, \bar{Y}_{N_L + N_U}$.
    Fine-tune the classifier $f_{t-1}$ with the pseudo labels which generates $f_t$.
**end for**

---

## 4  Experimental results

In this section, we first introduce some implementation details, and then present the experimental results on Cryo-EM dataset. Most of our implementations follow the setup in Fan et al. (2022). We briefly go through some important details, while referring the readers to the Appendix B for a complete setup.

**Dataset.** We experiment on a cryo-EM dataset called Y3 with 8653 holes over 9 grids, 58 squares, and 771 patches. We split the dataset into training and validation datasets with 6489 and 2164 holes respectively. Each hole corresponds to a state in the environment. The feature information for each state is the hole-level image observation. Note that the ground truth CTFs are valued by the micrographs, which can not be accessed through hole-level images.

**Perception Model.** We train a ResNet-18 (He et al., 2016) to classify the hole-level images for the perception model. Hyperparameters for *FixMatch* training is given in Appendix B.

**Input Features to DQN.** Apart from the hole-level predictions from the perception model, we add the following features to encode the geometric information for RL policy training. For each of the patch, square and grid, we compute the number of unvisited holes, unvisited low CTFs holes, visited holes and visited low CTFs holes within the patch, square and grid, respectively. Additionally, we have three dummy variables encoding whether the agent reaches a new patch, square or grid. During the training, the features of the past three steps are concatenated as the input of DQN. We terminate an episode whenever the duration, i.e. the cumulative sum of cost, reaches a certain threshold. Two thresholds 120 and 480 are considered for RL training. We use a three-layered MLP model with hidden sizes (128, 256, 128) and ReLU activation function for the Q-network.

### 4.1  Results

We experiment on 5%, 10% and 20% of the training data and conduct evaluation on the entire validation set. We compare our proposed approach with 3 baseline methods: a) the cryoRL method proposed in Fan et al. (2022) based on a supervised classifier (**SL**); b) cryoRL based on *FixMatch* (**FixMatch** ); and c) cryoRL based on **Iterative FixMatch** that runs the exact same iteration approach as $SSL^2$-*RL* except that the pseudo labels are given by the perception model itself instead of RL. We evaluate our proposed method for duration 120 and 480 minutes (i.e. **$SSL^2$-RL 120** and **$SSL^2$-RL 480**), respectively. For fairness, cryoRL is trained with both labeled and unlabeled data in all cases and evaluated at a duration of 480. The RL rewards and the accuracy of the corresponding perception models are presented in Table 1. For algorithms that do iteration, the best validation RL rewards and the corresponding accuracy are presented. For reference, when the fully labeled dataset is used, the classification model achieves an accuracy of 65.24%, and the best RL reward from cryoRL is 69.76. With 5% of the labeled data, *FixMatch* improves the classification accuracy by 5% compared with

supervised learning. By further increasing the labeled data to 10% and 20%, the improvement ($\sim 1\%$) becomes less obvious. Our proposed approach ($SSL^2 - RL480$) consistently outperforms FixMatch by $\sim 2\%$ and is on par with the supervised model trained using 100% labeled data (65.24%). As a comparison, iterative FixMatch performs only slightly better than FixMatch, clearly indicating the effectiveness of incorporating feedback from RL into perception learning. A similar trend can also be found in terms of the RL rewards (Table 1b), suggesting that RL benefit from improved classification as well. Iterative approaches performs in general better than the non-iterative approaches. Nevertheless, $SSL^2$-$RL$ 480 still outperforms *FixMatch* +iteration. Indeed, we observe that pseudo labels generated by the RL trajectories tends to have better precision score and general accuracy than the perception models at the previous round as shown in Figure 4 (c-e).

Table 1: A summary of RL rewards and classification accuracy of compared methods. Table (a) shows the classification accuracy for the perception model. For the iterative methods, we report the results that reaches the highest RL reward over 10 independent runs. Table (b) shows the average RL rewards and their standard deviation. Bold text marks the best RL rewards for each row.

(a) Classification accuracy

| % of labels | SL | FixMatch | Iterative FixMatch | $SSL^2$-RL 120 | $SSL^2$-RL 480 |
|---|---|---|---|---|---|
| 5% | 0.5707 | 0.6229 | 0.6372 | 0.6423 | **0.6451** |
| 10% | 0.6188 | 0.6303 | 0.6377 | 0.6480 | **0.6557** |
| 20% | 0.6299 | 0.6382 | 0.6396 | **0.6502** | 0.6479 |
| 100% | 0.6524 | - | - | - | - |

(b) RL rewards

| % of labels | SL | FixMatch | Iterative FixMatch | $SSL^2$-RL 120 | $SSL^2$-RL 480 |
|---|---|---|---|---|---|
| 5% | $59.55 \pm 5.4$ | $56.97 \pm 3.2$ | $62.33 \pm 7.5$ | $\mathbf{62.94} \pm 4.6$ | $61.62 \pm 7.1$ |
| 10% | $50.96 \pm 5.6$ | $58.50 \pm 5.5$ | $61.95 \pm 3.4$ | $64.28 \pm 8.5$ | $\mathbf{65.73} \pm 7.0$ |
| 20% | $56.76 \pm 7.3$ | $58.98 \pm 3.5$ | $65.77 \pm 4.2$ | $64.29 \pm 8.2$ | $\mathbf{67.28} \pm 6.3$ |
| 100% | $69.76 \pm 2.1$ | - | - | - | - |

Classification accuracy is not the best metric to reflect the RL performance. We further investigate the precision score metric. Precision measures the number of true positive out of all the positive samples predicted by the model. It is more consistent with the RL rewards since an episode is terminated at duration of 480 minutes, which only allows the agent to visit a small number of holes. In Figure 4 (a), we observe significant increases in the precision score during the iteration, which marks the overall improvement in the quality of the perception models, while the compared iteration method, *FixMatch* +iteration does not show a similar improvement in the precision score during the iteration. Another metric is to directly compare the number of low-CTF holes found by the trained RL agent. As shown in Figure 4 (b) SSL-RL has the dominant performance over other methods at different levels of percentages of labeled data.

## 5 Discussion and limitations

In this paper, we proposed $SSL^2$-$RL$ , an iterative framework that jointly learns the perception model and decision-making model. We focus on the navigation problem, which allows us to connect the learning of RL and that of the perception model by directly generating pseudo labels from trajectories. The framework demonstrates a significant improvement over a pure RL method Fan et al. (2022) on the cryo-EM data collection task. We then theoretically showed that RL with a penalty on large movement induces bias towards localization on the pseudo labels, which may improve the quality of the pseudo labels. A potential direction is to extend the framework to more general RL problems. Currently our approach only applies to navigation problem where the orders of visiting in a trajectory imply the labels for the perception learning. One potential way to generalize is to generate pseudo labels from the learned Q function.

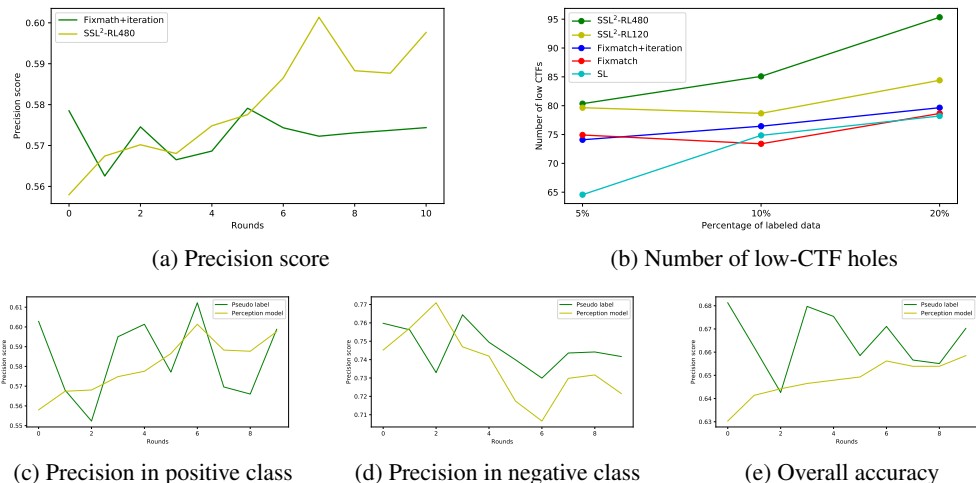

(a) Precision score      (b) Number of low-CTF holes

(c) Precision in positive class    (d) Precision in negative class    (e) Overall accuracy

Figure 4: (a) Changes in precision score over 10 rounds of iteration for *SSL$^2$-RL* 480 and *FixMatch* +iteration for 10% of the labeled data. (b) The average number of low CTF holes found by the trained RL agents within 480 duration for different methods under different percentages of labeled data. (c-e) The quality of pseudo labels compared to the perception models.

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

# A  Theoretical Understanding

In this section, we provide some theoretical insights into the benefits of our proposed method. A key to understanding our problem is whether RL could generate better pseudo labels than the classifier pretrained on the labeled dataset. Recall the pseudo label of the $i$-th state is denoted by $\bar{Y}_i$. We study whether $\sum_{i=1}^{N_U} \mathbb{1}(\bar{Y}_i = y_i) \geq \sum_{i=1}^{N_U} \mathbb{1}(f(x_i) = y_i)$.

**Benefits of RL label propagation.**    Pseudo labels from RL can be seen as a special way of doing label propagation. As opposed to label propagation through random walk, RL navigate on the map under the guide of the pretrained predictor. We understand the benefits of using RL for label propagation in the following two ways.

First, RL agents are trained with additional geometric information. It is normally not easy for a classifier to encode geometric information since most classifiers treats data i.i.d. For example, if the input features are images, popular image classification models does not directly incorporate dependence among images. Second, the movement costs added to the RL reward function induce bias towards localization of the true labels, which may align with the true label generating process. To this end, we use the example in Figure 5 to illustrate. The RL trajectory is able to correctly classify cluster 4 (on the right panel), because starting from 3 RL tries to avoid large movements.

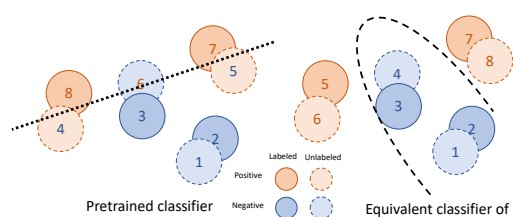

Figure 5: An illustration of localization bias from RL pseudo labels (adapted from Cai et al. (2021)). The black lines represent the (equivalent) decision boundaries of the pretrained classification model and RL. The numbers represents the visiting orders. The equivalent decision function by RL achieves 100% accuracy due to the localization.

**Localization of RL-based Label Propagation.** In this section, we rigorously discuss the localization property of the train RL policy. To this end, we introduce some extra setups. We assume that the marginal distribution of $(s, x)$ is $L$ for the labeled dataset and $U$ for the unlabeled dataset. Let the inconsistency rate between two predictors $g_1, g_2 : \mathcal{S} \times \mathcal{X} \mapsto \mathcal{Y}$ be $\mathcal{E}^L(g_1, g_2) = \mathbb{E}_{s,x \sim L} \mathbb{1}(g_1(s, x) \neq g_1(s, x))$.

In the literature, label propagation is given by regularizing the consistency across neighboring points. Cai et al. (2021) proposes to solve the following optimization problem for a improved classifier $f^*$:

$$f^* = \operatorname*{argmin}_{f:\mathcal{S}\times\mathcal{X}\to\mathcal{Y}} \mathcal{E}^L\left(f, f_{tc}\right) \text{ s.t. } R_\mathcal{B}(g) \leq \mu, \text{ for some } \mu > 0, \tag{2}$$

where $f_{tc}$ is the pretrained classifier and the regularization is defined by

$$R_\mathcal{B}(f) = \mathbb{P}_{s,x\sim\frac{1}{2}(L+U)}\left[\exists s' \in \mathcal{B}(s), \text{ s.t. } f(s', x) \neq f(s, x)\right],$$

and $\mathcal{B}(s)$ is the neighboring of $s$. In practice, one optimizes its empirical version.

Now we define the object of RL training. For a trained policy $\pi$, let $(S_1, Y_1, \ldots, S_{N_U+N_L}, Y_{N_U+N_L})$ be the trajectory of visited state and labels by evaluating $\pi$ on the whole dataset. We aim at finding the policy $\pi$ that maximizes the regularized cumulative rewards up to step $N_C$:

$$\pi^* = \operatorname*{argmax}_\pi \sum_{t=1}^{N_U+N_L} \mathbb{1}(Y_t = 1)\mathbb{1}(t \leq N_C) - c(S_t, S_{t+1})^2. \tag{3}$$

Slightly abusing the notation, we let the equivalent decision boundary of a policy $\pi$ by $f_\pi : \mathcal{S} \times \mathcal{X} \mapsto \{0, 1\}$, such that $f_\pi(s, x) = \mathbb{1}(t(s, \pi) \leq N_C)$, where $t(s, \pi)$ is the step in which $s$ being visited by running policy $\pi$. We have the following lemma that proves the equivalence between the two regularization.

---

[2]Note that we consider the policy running through the whole dataset even after it is terminated

**Lemma 1.** *Let* $\{\mathcal{S}_1 \ldots, \mathcal{S}_B\}$ *be a* $B$-*partition of* $\mathcal{S}$, *i.e.* $\cup_{b=1}^B \mathcal{S}_b = \mathcal{S}$ *and let* $P(s)$ *be the partition* $s$ *belongs to. We define the neighbor function by the partitions, i.e.* $\mathcal{B}(s) = \{s' \in \mathcal{S} : P(s') = P(s)\}$. *Then for all policy* $\pi$, $\hat{\mathcal{R}}_{\mathcal{B}}(f_\pi) \leq C_1 \sum_t c(S_t, S_{t+1}) + C_2$, *for some universal constants* $C_1$ *and* $C_2$. *The equality holds, if* $\pi$ *visits each partition at most twice.*

*Proof.* We first rewrite the empirical regularization term:

$$\hat{\mathcal{R}}_{\mathcal{B}}(f_\pi) = \sum_{b=1}^B n_b \mathbf{1}(\exists s_1, s_2 \in P_b, (\pi(s_1) - N_C) \times (\pi(s_2) - N_C) < 0).$$

The cumulative penalty term are the total number of switches between partitions. For each partition, whenever $\exists s_1, s_2 \in P_b, (\pi(s_1) - N_C) \times (\pi(s_2) - N_C) < 0$, an extra switch is introduced. Thus we have the first inequality:

$$\sum_{i=1}^{N_L + N_U} c(S_i, S_{i+1}) \geq B + \sum_{b=1}^B \mathbf{1}(\exists s_1, s_2 \in P_b, (\pi(s_1) - N_C) \times (\pi(s_2) - N_C) < 0)$$

$$\geq B + \frac{1}{\min_b n_b} \hat{\mathcal{R}}_{\mathcal{B}}(\pi).$$

The equality can be achieved when $\pi$ visits the states in a cluster all at once unless some of them have different labels. $\qquad\square$

Cai et al. (2021) shows that the improved classifier can achieve arbitrarily small classification error even if the error rate of the pretrained classifier is high, which shows a potential benefits of regularizing the inconsistency across samples that are closed. Lemma 1 indicates that the similar improvement can be expected for RL based label propagation.

# B   Training Details

## B.1   Details on reward function

We assign a positive reward 1.0 to the agent if an action results in a target hole with a CTF value less than 6.0Å and 0.0 otherwise. The agent also receives a negative reward depending on the operational cost associated with a hole visit. Let $\mathcal{P}_s, \mathcal{Q}_s, \mathcal{G}_s$ be the patch, square and grid index of the hole $s$. In the end, all the possible rewards and their corresponding conditions are given by

$$r(s_i, a_i) = \begin{cases} 1.0 & \text{if } \text{ctf}(s_{i+1}) < 6.0 \ \& \ \mathcal{P}_{s_i} = \mathcal{P}_{s_{i+1}} \\ 0.57 & \text{if } \text{ctf}(s_{i+1}) < 6.0 \ \& \ \mathcal{P}_{s_i} \neq \mathcal{P}_{s_{i+1}} \ \& \ \mathcal{Q}_{s_i} = \mathcal{Q}_{s_{i+1}} \\ 0.23 & \text{if } \text{ctf}(s_{i+1}) < 6.0 \ \& \ \mathcal{Q}_{s_i} \neq \mathcal{Q}_{s_{i+1}} \ \& \ \mathcal{G}_{s_i} = \mathcal{G}_{s_{i+1}} \\ 0.09 & \text{if } \text{ctf}(s_{i+1}) < 6.0 \ \& \ \mathcal{G}_{s_i} \neq \mathcal{G}_{s_{i+1}} \\ 0.0 & \text{otherwise} \end{cases}$$

where $s_{i+1} = T(s_i, a_i)$.

## B.2   Hyperparameters

There are three hyperparameters for the training of *FixMatch* . **uratio** controls the ratio between the number of samples from labeled data and the number of samples from unlabeled data in each batch. **ulb_loss_ratio** is the coefficient of the unsupervised loss. The two hyperparameters are set to 4 and 5.0 respectively. **p_cutoff** ($\tau$ in 1) controls minimum confidence it requires to be considered for the unsupervised loss, which is set to 0.8.

The initial classifier is a Resnet 18, trained under learning rate 0.01 with a cosine learning rate scheduler, dropout rate 0.5, batch size 64 for 200 episodes. The in-loop fine-tuning is trained with a learning rate of 0.001 for 40 episodes.

The Reinforcement Learning model is trained with learning rate 0.001 and 500 steps per epoch for 10 epochs.

# C Additional Experimental Results

Though our problem is a binary classification problem, the target labels CTF are extremely noisy. From Figure 6, we can see that many samples lie around the threshold 6, which is used to decide high and low CTFs in this paper.

## C.1 Dataset

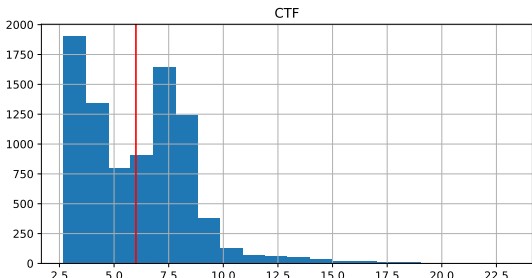

Figure 6: Histogram of the CTF scores over the whole dataset.

## C.2 Model selection based on accuracy

In Table 1, we select model for iteration approaches based on their validation RL rewards. In this section, we compare the model selected by the best accuracy. The results remains the same for most of the cells.

Table 2: A summary of RL rewards and classification accuracy of compared methods. Table (a) shows the average RL rewards and their standard deviation for different methods under 5%, 10%, 20% and 100% of labeled training dataset. Bold text marks the best RL rewards for each row. Table (b) shows the classification accuracy for the perception model. For the iterative methods, we report the results that reaches the highest RL reward over 10 independent runs.

(a) RL rewards

| % of labels | SL | FixMatch | FixMatch +iteration | $SSL^2$-RL 120 | $SSL^2$-RL 480 |
|---|---|---|---|---|---|
| 5% | $59.55 \pm 5.4$ | $56.97 \pm 3.2$ | $\mathbf{62.33 \pm 7.5}$ | $61.75 \pm 6.9$ | $61.62 \pm 7.1$ |
| 10% | $50.96 \pm 5.6$ | $58.50 \pm 5.5$ | $59.32 \pm 2.6$ | $64.28 \pm 8.5$ | $\mathbf{65.73 \pm 7.0}$ |
| 20% | $56.76 \pm 7.3$ | $58.98 \pm 3.5$ | $65.77 \pm 4.2$ | $64.29 \pm 8.2$ | $\mathbf{67.28 \pm 6.3}$ |
| 100% | $69.76 \pm 2.1$ | - | - | - | - |

(b) Classification accuracy

| % of labels | SL | FixMatch | FixMatch +iteration | $SSL^2$-RL 120 | $SSL^2$-RL 480 |
|---|---|---|---|---|---|
| 5% | 0.5707 | 0.6229 | 0.6372 | 0.646 | **0.6451** |
| 10% | 0.6188 | 0.6303 | 0.653 | 0.6480 | **0.6557** |
| 20% | 0.6299 | 0.6382 | 0.6396 | **0.6502** | 0.6479 |
| 100% | 0.6524 | - | - | - | - |

## C.3 ANOVA test

We run ANOVA test on the reduction of sum of squares of CTF scores at different magnification levels.

Table 3: ANOVA test on different magnification levels

| Levels | Sum of Square | Cumulative | df | p-value |
|--------|---------------|------------|------|---------|
| Grid   | 5393.9        | 116737     | 9    | 1.1e-16 |
| Square | 17167.2       | 111343     | 58   | 1.1e-16 |
| Patch  | 31570.4       | 94175      | 771  | 1.1e-16 |
| Hole   | 62604         | 62604      | 5997 | -       |

## C.4 Ablation study

In this section, we conduct experiments to characterize the proposed approach. We investigate the following components including the choice of termination strategies, the importance of using semi-supervised learning for RL, the essential of using cost penalty inside the iteration and the performance of other RL methods.

**Termination strategy.** In table 1, we compare the results of iteration approaches that terminate when the RL reward is the highest. One can also terminate when classification accuracy reaches the highest. The results are given in Appendix C Table 2, which is similar to Table 1.

**Without Semi-supervised RL.** We use only 10% of the labeled data to train CryoRL policies in a supervised way. Figure 7 (a) shows the change of RL rewards during a 10-round iteration. There is a significant gap between semi-supervised RL and supervised RL that trains only on the 10% labeled data. This strongly suggests that unlabeled data is beneficial for RL.

**Without moving cost penalty.** Though we will show later that the movement cost introduces a strong bias towards localization, which may improve the quality of pseudo labels, we empirically investigate the benefits of adding the movement penalty. We can see that the classification accuracy does not increase. Figure 7 (b) shows the change of classification accuracy of $SSL^2$-$RL$ 480 for a 10-round iteration on 10% data. We don't see a significant increase in classification accuracy. It also performs worse than the results reported in Table 1.

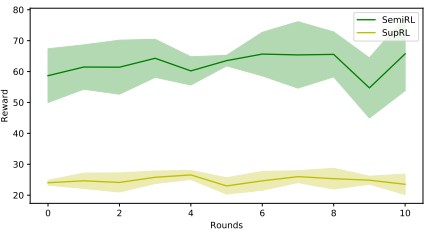
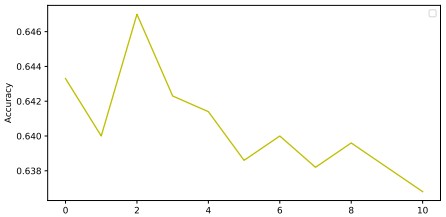

(a) Semi-supervised RL vs. supervised RL      (b) Acc. for $SSL^2$-$RL$ 480 without movement cost

Figure 7: (a) RL rewards for 10-rounds $SSL^2$-$RL$ 480 with policies trained by semi-supervised RL and supervised RL, respectively. (b) Classification accuracy for 10-round $SSL^2$-$RL$ 480 without movement penalty on 10% of labeled data.

**Other RL models.** DQN is used for decision-models in Table 1. We replace DQN with other RL models, e.g. A2C (Rosenstein et al., 2004) and Rainbow (Hessel et al., 2018). Both A2C and Rainbow are worse than $SSL^2$-$RL$ 480 using DQN, which is consistent with the observations in Fan et al. (2022).

Table 4: Performances of Rainbow and A2C compared with DQN

| Metrics | $SSL^2$-RL 120 | $SSL^2$-RL 480 | Rainbow 480 | A2C 480 |
|---------|----------------|----------------|-------------|---------|
| Accuracy | 0.6557 | 0.6480 | 0.6400 | 0.6430 |
| RL rewards | 64.28+9.5 | 65.73+7.0 | 62.80+3.1 | 61.80+5.7 |

