# OpenReview forum: "Coupling Semi-supervised Learning with Reinforcement Learning for Better Decision Making --- An application to Cryo-EM Data Collection"
_NeurIPS.cc/2023/Workshop/AI4Science — NeurIPS2023-AI4Science Poster_

### Official Review · Reviewer_QtFv · 2023-10-25
**review from reviewer QtFv**

**Rating:** 5
**Confidence:** 3

**Review:**

This paper targets the problem of seeking maximum high-quality micrographs taken by cryo-electron microscopy via navigating at different magnification levels. The authors propose an iterative framework that alternates perception modeling and RL policy learning for coupling and improving training simultaneously. Specifically, under this semi-supervised RL framework, the error from the perception model with pseudo-labels can be corrected.  Their proposed method outperforms various baseline methods in terms of both RL rewards and the accuracy of the perception model.

## Pros
1. The integration of the perception model and RL looks reasonable and the iterative framework is able to improve both of them gradually.

## Cons
1. The improvement of the proposed method compared with the baseline is actually not significant in Table 1.
2. In Table 1, $SSL2-RL 480$ has a better accuracy with 10% compared with 20% while it has a higher reward with 20% compared with 10%. I was wondering if there is any mismatching for the reward function.

## Question
1. In practice, is it hard to make sure that the learning can converge eventually? It seems that it still depends on the difficulty of the problem.
2. It will be interesting if authors can also fill up the entries of 100% in Table 1 to see if the proposed method can still be better than the baseline.
3. Is there any insight, reference, or ablation study why selecting the duration of 120 and 480?

---

### Meta-Review · Area_Chair_3D3R · 2023-10-27

**Recommendation:** Accept (Poster)
**Confidence:** 4

**Metareview:**

**Summary:**
The paper presents an innovative iterative framework that amalgamates semi-supervised Reinforcement Learning (RL) with a perception model. The research is notably applied to cryo-electron microscopy (cryo-EM) data collection, emphasizing the acquisition of high-quality micrographs through navigation at varied magnification levels.

**Strengths:**

1. **Novelty and Relevance:** The research addresses a significant challenge in the RL domain, where the efficacy of an RL agent can be curtailed by the quality of the perception model, especially in situations with scarce labeled data. The iterative framework proposed is both unique and timely, catering to this challenge.

2. **Practical Application:** Application to cryo-EM data collection is compelling, given the importance of obtaining high-quality micrographs in scientific research. This grounds the theoretical work in a tangible, real-world scenario.

3. **Performance Metrics:** The research underscores its value by contrasting its performance against various baseline methodologies. It's commendable that the proposed method trumps the competition in both RL rewards and perception model accuracy.

4. **Theoretical Insights:** The paper doesn't just stop at empirical validation; it ventures into offering theoretical insights. The elucidation of RL-generated pseudo labels' bias towards localization is insightful and fortifies the credibility of the approach.

**Areas of Improvement:**

1. **Generalization:** While the application to cryo-EM is pivotal, the paper could benefit from showcasing the versatility of the proposed framework in other domains or applications, ensuring readers of its broad applicability.

2. **Assumption Clarification:** It's essential to comprehend the assumptions made while building the iterative framework. For instance, the belief that the decision model can correct the perception model's errors is foundational. A deeper dive into this assumption's validity and potential limitations would enhance the paper's robustness.

3. **Complexity Analysis:** While the performance metrics are compelling, it would be helpful to understand the computational cost or complexity associated with the iterative framework, especially in comparison to the baselines.

**Conclusion:**
The paper presents a groundbreaking approach in coupling semi-supervised learning with RL, particularly for situations marked by limited labeled data. Its real-world application, empirical achievements, and theoretical insights collectively underscore its value in the domain of machine learning. However, a deeper exploration into its broad applicability, underlying assumptions, and computational efficiency can further enhance its impact and relevance.